# Assessment of heavy metals in soil and water from Bahi district, Tanzania

**Dominic Parmena Sumary**[1,2]*, **Jofrey Raymond**[1], **Musa Chacha**[1], **Firmi Paul Banzi**[3], **Edwin Gomezulu**[2]

**1** Department of Global Health and Bio-Medical Sciences, Nelson Mandela African Institution of Science and Technology, Arusha, Tanzania, **2** St John's University of Tanzania, Dodoma, Tanzania, **3** Tanzania Atomic Energy Commission, Arusha, Tanzania

\* dsumary@gmail.com

**Editor:** Mohamed Y.M. Hanfi, Ural Federal University named after the first President of Russia B N Yeltsin: Ural'skij federal'nyj universitet imeni pervogo Prezidenta Rossii B N El'cina, RUSSIAN FEDERATION

## Abstract

This study presents analysis of heavy metals concentrations in soil and water samples collected from villages near Bahi Swamp found in Bahi district, Tanzania. The research involved quantitative analyses methods of heavy metals in the laboratory and the use of descriptive statistical methods to draw conclusion. A total of 45 soil samples and 21 water samples were collected from five locations near Bahi Swamp which was pre-selected from northern and the eastern part. Heavy metals from soil and water samples were analyzed by using energy dispersive X-ray fluorescence (ED-XRF) spectrometry and atomic absorption spectrometry (AAS) technique, respectively. The concentrations of nine heavy metals (Pb, Cd, Mn, Zr, Cu, As, Zn, Sr, Cs) in soil samples and three in water samples (Mn, Zn, Cu) were determined. The mean concentrations of heavy metal from soil samples were Pb ($29.60 \pm 7.50$ mg kg$^{-1}$), Cd ($403.20 \pm 507.44$ mg kg$^{-1}$), Mn ($478.27 \pm 245.86$ mg kg$^{-1}$), Zr ($206.00 \pm 79.47$ mg kg$^{-1}$), Cu ($52.00 \pm 5.24$ mg kg$^{-1}$), As ($5.27 \pm 1.66$ mg kg$^{-1}$), Zn ($48.47 \pm 31.18$ mg kg$^{-1}$), Sr ($21.93 \pm 36.99$ mg kg$^{-1}$), and Cs ($34.00 \pm 10.95$ mg kg$^{-1}$). Soil samples exhibited diverse trace element patterns, with concentrations following the order As < Pb < Zn < Cu < Sr < Cs < Zr < Cd < Mn. Notably, Cd concentrations in Bahi Sokoni (BSS) and Bahi Matajila (BTS) exceeded other locations by over 20 times of the concentrations ranged from 38 to 1157 mg kg$^{-1}$. Manganese (Mn) concentrations varied significantly, but were still within the permissible limits set by regulatory authorities such as, FAO/WHO, US EPA and TBS. Also, the study of metal oxides found that the mean concentration of tellurium oxide (TeO$_2$) in soil samples was significantly higher than typical natural abundance levels. Water samples displayed a wide range of Mn concentrations, while Zn was detected in specific sites. Thus, the findings pose a potential health and environmental concerns, serving as contribution for further research and highlighting environmental regulatory compliance with an emphasis that Bahi is known for its uranium deposits.

**Data availability statement:** All relevant data are within the manuscript and its Supporting Information files.

**Funding:** This work is partly funded by the St John's University of Tanzania under the scheme of staff development, for PhD candidate.

**Competing interests:** The authors have declared that no competing interests exist.

## Introduction

Bahi district, located in central Tanzania, is underlain by Late Archean rocks of the Tanzaman shield, predominantly biotite granites along with metasediments like migmatites and gneisses. This geological and geochemical characteristic support the presence of uranium mineralization which highly suggests geo-ecological with other toxic elements. Globally, naturally occurring heavy metals are widespread in our environment, primarily due to geographical factors. Numerous studies have been conducted and reported worldwide on sediments with respect to both trace elements, essential elements not only due to its paramount importance, but on its detrimental effects to both animals and plants bodies [1–4]. Most heavy metals which occur naturally in the environment are associated with sediments, underlying rocks, and water [5]. The circulation and migration of heavy metals in the environment are primarily driven by process such as rock weathering, volcano eruptions, surface water runoff, and wind [6,7]. However, the heavy metal contamination is largely contributed by anthropogenic activities which relates to industrial waste discharge, mining, agricultural chemicals, and other organic or inorganic by-products [8]. Unrestricted and illegitimate of these anthropogenic activities involving heavy metals usually lead to environmental pollution that becomes prone to deadly life-threatening hazards to human health and the ecosystem [9]. Globally, in years between 1980 and 2000 the main sources of metal pollution in rivers and lakes were mining and manufacturing, along with waste discharge, which had a combined contribution of ~90% [10,11].

Heavy metals contamination and exposure are results of various factors that form a complex matrix of items. Heavy metals have been associated with health risks at low concentrations, mostly above tolerable limit range from $1–1000\,mg\,kg^{-1}$ [12]. Certain trace elements can pose health risks, leading to various adverse effects, for instance: lead (Pb) which is linked to developmental and cognitive issues, cadmium (Cd) is associated with kidney damage and an increased risk of cancer, manganese (Mn) may impact neurological systems, copper (Cu) and zinc (Zn) may lead to gastrointestinal complications and antagonistic effects, arsenic (As) is linked to health problems associated with skin, lung, and bladder cancers, strontium (Sr) can cause hypocalcemia (low calcium levels), and cesium (Cs) poses a radioactivity risk [10,13–16].

In conjunction with heavy metals, the understanding of chemical ecology through the study of metal oxides distribution have been important since they play a pivotal role in the soil by shaping the physical, chemical, and environmental characteristics of terrestrial ecosystems [17,18]. These compounds are essential components of the Earth's crust and arise from the weathering of parent rock materials, influencing soil properties and, consequently, plant health bringing a natural balance in the overall ecosystem [19]. Metal oxides, including iron ($Fe_2O_3$), silicon ($SiO_2$), aluminum ($Al_2O_3$), calcium (CaO), titanium ($TiO_2$), and barium (BaO), exhibit diverse concentrations in the soil, reflecting their geochemical origins and contributions to nutrient availability, heavy metal mobility, and plant stress responses [20,21]. Understanding the distribution, interactions, and environmental implications of metal oxides in soil is fundamental for sustainable land management, environmental conservation and health risks management [22,23].

The increase in health risks caused by high levels of heavy metals depends on how these metals move from the environment to plants and animals. Health concerns from heavy metal contamination from the environment to humans can be through multiple routes of exposure, including inhalation, dermal absorption, or ingestion, particularly through the soil-food chain interaction [13,16]. In the context, the indicators of the severity of health impacts are most prominently evident in animals, of which humans are in a focus.

The geological features of Bahi area are characterized by the underlain Late Archean rocks of the Tanzaman shield, 2550 million years, mainly comprised of biotite granites, together with metasediments (of the type migmatites, gneisses, quartzites, amphibolites), and greenstones which are metavolcanics of the Nyanzian System. The unaltered granites usually contain less than 6 ppm $U_3O_8$, which occur in allanite, sphene, zircon and apatite and are, therefore, *a priori* not fertile. Thus, uranium minerals could be determined only in the highly mineralized samples of the pedogenic mbuga calcretes and silicified sheet calcretes. Spotty uranium mineralization has been found in strongly silicified pedogenic sheet calcretes (0.1 to 0.5 m thick) overlying weathered granitic rocks and exposed in creek beds at Kisalalo (Bahi North area) and at Bahi East. Grab samples contain 75–470 ppm $U_3O_8$ but less than 100 ppm $V_2O_5$. At Kisalalo, the white, porous, silicified calcrete (46.9% $SiO_2$, 30.9% CaO) contains spotty visible secondary uranium mineralization in the form of weeksite. It occurs in 0.1 mm long flakes coating opal-like silica that forms concretionary textures and irregular bands replacing the extremely fine-grained calcitic ground mass [24].

Although there have been various reports on the distribution and impacts of heavy metals in various environments, there is limited research focusing specifically on Bahi district that delve into levels, distributions, and association with its geochemical ecology. Therefore, the main motive and objective of this research was to assess concentration of heavy metals from the terrestrial environments; soils and water samples from the selected sites in Bahi district due to its geological and geochemical characteristics dominated by uranium deposits. Thus, the present study aimed at determining the concentration of heavy metals and metal oxides in Bahi's terrestrial environment, evaluating their distribution and their association in geochemical ecology, and assessing health risks based on internationally recognized standards. In this regard, by understanding its levels then the information can potentially be used to foresee the health risks that may be associated, since both heavy metals and radionuclides lead to similar health challenges.

## Materials and methods

### Study area

The present study was conducted at Bahi district in Dodoma region, Tanzania. Bahi district which is among the seven districts of Dodoma region found at 5°58'24.5"S and 35°19'23.45"E, covering 5 630 km2. The district is located about 56 km west of Dodoma City. It is bordered to the North by Chemba district, to the East by Dodoma and Chamwino districts (which covers North-East and South), and to the West by Singida Region (Fig 1). Regional-wise, total rainfall ranges from 500 mm to 800 mm per annum with high geographical, seasonal and annual variation with a mean monthly temperature of 22.6 °C [25–27]. According to the census report of 2022, Bahi district comprised a population of 322,526 people, with 75792 households at density 57.3/ $km^2$ of which 95.8% is rural area [28,29]. About 59% of households in Bahi district are engaged in crop farming with which 29% of the total planted area was under irrigation whose main water source was from rivers. The total planted area is comprised of cereals 40499 ha, roots & tubers 70 ha, oil seeds & oil nuts 23436 ha, and fruits & vegetables 141 ha [30]. Significantly, Bahi economy is famous characterized by Lake Sulunga (Bahi Swamp) having surface area of approximately 974 $km^2$ located within the Bahi depression, 830 meters above sea level. The lake lacks an outflow; hence its size fluctuates significantly based on precipitation in the catchment area, and it occasionally dries up completely in some years [31,32]. The geographical and geological landscape near Lake Sulunga, which extends between Dodoma and Singida region comprises a wetland area covering approximately 1250 $km^2$ [32].

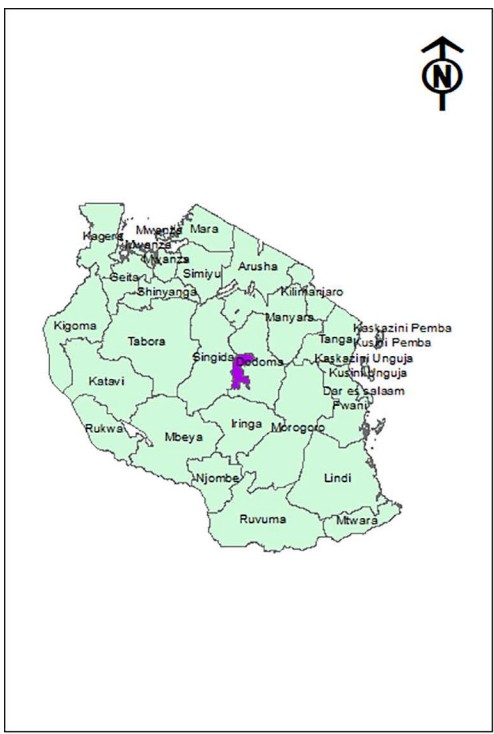

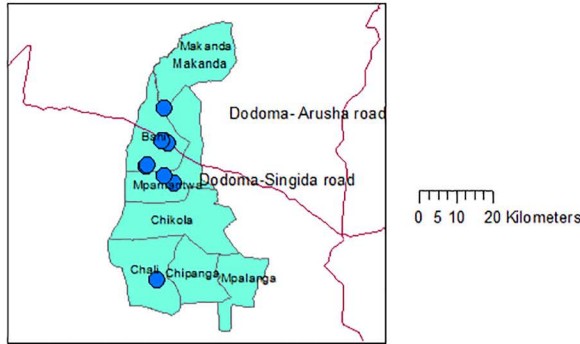

**Fig 1. The map of the study area showing sampling points.**

## Permits and ethical considerations

There was no specific governmental permit numbers required for conducting fieldwork in the study area. However, to ensure transparency and community awareness regarding the research activities, appropriate institutional and local authorization were obtained. After the review and approval of the research proposal by the Nelson Mandela African Institution of Science and Technology (NM-AIST), the authorization was granted virtually to continue with data collection and experimentation. In addition, an introduction letter was issued by St John's University of Tanzania to the Bahi District Executive Director. This official who is responsible for overseeing local governance and the implementation of the district's strategic plans, facilitated the researchers' access research site by approving the letter to proceed to relevant ward leaders to contact the landowners. Thus, soil and water samples were collected with knowledge and verbal consent of these local stakeholders.

## Apparatuses and instruments

The handheld GPS (model: DAKOTA 20 Taiwan, FCC ID: IpH01543) was used to record the geographic coordinates of the sampling locations to enable accurate mapping and analysis of spatial data. Concentrations of heavy metals in the homogenized soil samples were determined by Niton XL3t Energy Dispersive X-ray Fluorescence (ED-XRF) analyzer (Thermo Fisher Scientific, Australia) available at Geological Survey of Tanzania (GST). Durable and contamination-free polyethylene bags about 30 μm in thickness were used to keep and transport soil samples. A hot air oven (model: Lattech®, 50 Hz, made in India) was used to dry the sample after being dried in air. The pulverizer was used to grind the soil samples. Water samples were stored and transported in a well-sealed polyethylene bottle. A digital pH and conductivity meter (model: Hanna Instruments HI 98107 pH Meter) was used on site for these parameters. Mn, Zn, and Cu were analyzed in each digested sample solutions using an Air/Acetylene-Varian SpectrAA-55B Flame Atomic Absorption Spectrophotometer (FAAS) (Varian Inc., Australia) available at GST laboratory, Tanzania.

## Materials and chemicals

A shovel, a wooden pike, and a trowel were used in digging, and transfer of soil samples into containers to prevent contamination. Filter papers, Whatman No. 42 were used to remove particulate matter from water samples. The chemicals and reagents used for laboratory analyses were all of the analytical grades such as $HNO_3$ (65% w/w, 1.42 g mL$^{-1}$), HCl (37% w/w, 1.18 g mL$^{-1}$) and $HClO_4$ (70% w/w, 1.67 g mL$^{-1}$) 1000 mg L$^{-1}$ stock standard solutions of heavy metals (Mn, Zn, and Cu) were purchased from Sigma-Aldrich Co. Ltd., UK. Deionized water from Geological Survey of Tanzania (GST). Two certified standard reference soils were used in this study for investigating the accuracy, precision and repeatability of the XRF method (NIST SRM 2710, Montana I and NIST SRM 2711a Montana II, soils for highly and moderately elevated trace element concentrations, respectively.

## Sample coding and recording

For laboratory analysis records, sample identity (ID), GPS coordinates of sampling locations for soils and water were recorded. The sampling techniques involved both random sampling and stratified sampling due to variation of samples and sampling location. The *ALSMatriXGpDN* sampling tool, is a coding tool designed by the researcher as presented in Table A and B for soil and water samples, respectively (S1 Table), to help record and organize field sample data systematically in accordance with other general recommended sampling methods [33].

## Collection and preparation of soil samples

A total of 45 soil samples from the selected sites were collected during the dry season in June/July to obtain maximal heavy metals. The five study sites namely; Nagulo, Makulu, Sokoni, Matajila, and Chang'ombe. In each location, three farm fields were selected randomly (at least 100 m apart) where soil samples were collected at three different points in each farm to represent the entire area [34,35]. Up to −15 cm of soil in the upper depth layer was sampled. Every soil sample weighing about 2 kg was put in a prepared polythene bag with a code-label [14]. All sampling points were georeferenced using a handheld GPS receiver. This was crucial so that the locations are identified, also to enable the preparation of maps showing the concentration range in different sampling points by using ArcMap software. The acquired GPS [33] data were converted to longitude and latitude readings in-line with the mean concentrations for each heavy metal analyzed in a given sampling location. After the collection of all samples, they were transported to the laboratory at St John's University of Tanzania for initial preparation and storage, later they were transported to Geological Survey of Tanzania (GST) laboratory for analysis. The samples were cleared of stones and pebbles then dried in an oven at a temperature of 105 °C for 4 hours to allow complete dryness. The soil samples were ground into fine particles, thoroughly mixed and passed through fine mersh sieve (2 mm) to obtain composite representative sample and later packed into labeled 500 mL containers.

## Collection and preparation of water samples

A total of 21 water samples were conveniently collected from seven (7) water sources used for domestic and agricultural purposes. These include, surface water of the Bubu river, 3 samples from domestic water wells in Bahi Sokoni, Bahi Town, Chali town and other 3 water samples were collected from Bahi Makulu, Bahi Swamp and a nearby shallow well [36]. Water samples were collected in triplicate and filled into pre-cleaned labeled (coded) polyethylene containers and added 2 mL nitric acid to prevent bacterial activities and adsorption of heavy metals on the container wall. Water samples were filtered and digested according to the standard methods for the examination of water according to USEPA method 200.9 and 3005 [37].

## Laboratory analysis of heavy metals and metal oxides

**Analysis of soil samples by ED-XRF.** An energy-dispersive X-ray fluorescence (ED-XRF) technique was used to determine the composition of selected trace elements in soil samples. The method is highly regarded for its non-destructive, sensitivity and precise quantification of trace elements. To ensure accuracy, the instrument's software was calibrated by using a pellet standard sample to determine the elemental compositions and concentrations. The concentrations of Pb, Cd, Mn, Zr, Cu, As, Zn, Cr, and Cs in the homogenized soil samples were determined. 8 g of each ground-sieved soil sample was weighed and placed in a polyethylene XRF cup (~ 30 mm) with double open rings and polypropylene XRF film at the base. 15 g of powdered samples were pressed into pellets without binder. After pressing the soil samples, the capped sample cups were placed in the ED-XRF shroud one at a time and scanned for 3 minutes of which each sample was analyzed in triplicate. The samples were then scanned with XRF mounted with a large-area high-throughput silicon drift detector (SDD). The heavy metals and metal oxide concentrations from the ED-XRF analysis were obtained in parts per million (ppm) by setting the detection limit of 0.1 ppm for heavy metals and tellurium, whereas metal oxides in which the limits were set at 0.01 ppm [38,39].

**Analysis of water sample by AAS.** A wet digestion flame atomic absorption spectrometry (AAS) technique was used for water samples by following standard procedures [40,41]. The EPA vigorous digestion method described by Gregg (1989) was adopted. 100 mL of each of representative water samples were transferred into Pyrex beakers containing 10 mL of conc. $HNO_3$. The samples were boiled slowly and then evaporated on a hot plate to the lowest possible volume (about 20 mL). The beakers were allowed to cool and another 5 mL of conc. Nitric acid was added. The heating was continued with addition of conc. nitric acid as necessary until digestion was complete. The samples were evaporated again to dryness (but not baked) and the beakers were cooled, followed by the addition of 5 mL of HCl solution (1:1 v/v). The solutions were warmed and 5 mL of 5M NaOH was added, then filtered. The filtrates were transferred to 100 mL volumetric flasks and diluted to the mark with distilled water. These solutions were then used for the elemental analysis. The instrument was calibrated according to the manufacturer's instructions before it was used. Mn, Zn, and Cu were analyzed in each digested sample solutions. Prior to experimentation, the instrument was calibrated using freshly prepared 1000 mg $L^{-1}$ standard stock solution of each respective metal as presented in Table C and D (S1 Table). All samples were examined in triplicates to ascertain the reproducibility of the results. The detection limits of all selected heavy metals were at 0.01 mg $L^{-1}$.

## Quality control of heavy metal analysis

To assess the accuracy provided by the ED – XRF technique, the Montana soil 2711A Standard Reference Material (SRM) obtained from the National Institute of Standards and Technology (NIST) along with a high-purity silica blank (99.8% $SiO_2$, supplied by Sigma-Aldrich Co. Ltd., UK) were utilized. The mean recovery rates for SRM analyses ranged between 89 ± 1.6% and 98 ± 2.6%, with reproducibility reflected in relative standard deviations (RSDs) of 3.4–4.5%. These results are consistent with the accepted control limits of 80–120% for SRM recoveries and an RSD threshold of <15% for metal analysis as presented in Table E and F (S1 Table). Furthermore, blank sample assessments confirmed the absence

of background contamination. The homogeneity of the analyzed samples was achieved by the preparation and used a starch binder throughout the entire analytical process. It was crucial to carry out the assessment in order to identify any notable contamination or interference that might have influenced the reporting of elevated metal concentrations during the analysis [42]. The analytical findings pertaining to the binder material did not show any significant contamination or interference in relation to each analyte; Pb, Cd, Mn, Zr, Cu, As, Zn, Cr, Cs and the metal oxides. Together with the technical laboratory standards and standard operation procedures ISO/IEC 17025:2017 adopted by GST were followed [43]. The results obtained were compared and discussed in comparison with the permissible limits (Tables 1 and 2).

The precision and accuracy of an AAS results were assessed using both blanks and SRM (NCS CZ 730310) from Beijing, China. Analyses of blank samples indicated no detectable contamination from reagents or glassware, with levels consistently below the detection limit (BDL) (0.01 mg L$^{-1}$). Three replicates of the NCS CZ 730310 material underwent the full digestion process, yielding average recoveries between 92.8% and 99.8%, with RSDs ranging from 5–11%. These results align with the permissible criteria of <15% RSD for metal analysis and recovery rates within the 80–120% range for SRMs [59].

## Statistical analyses

The statistical analysis of heavy metals concentrations from different locations were performed using analysis of variance (ANOVA), with the computations being performed with the software OriginPro (OriginLab Corporation, version 2023b, 64

**Table 1. Maximum standard permissible heavy metal concentrations (mg kg$^{-1}$) in soil.**

| Organization | Pb | Cd | Mn | Cu | Zn | As | Cr | Reference |
|---|---|---|---|---|---|---|---|---|
| EU | 300 | 3 | NI | 140 | 300 | 20 | 150 | [44–47] |
| WHO | 100 | | NI | 100 | NI | NI | NI | [47,48] |
| US. EPA | 200 | 0.43 | – | NI | NI | NI | NI | [49] |
| TBS | 200 | 1 | 1.50E + 03 | 200 | 150 | 1 | NI | [50] |
| FAO/WHO | 100 | 3 | 1.00E + 05 | 100 | 300 | NI | 100 | [51] |
| Canada | 200 | 3 | NI | NI | 250 | 20 | NI | [52] |
| South Africa | 20 | 7.5 | NI | NI | 240 | 5.8 | NI | [53] |
| China | 80 | 0.5 | NI | NI | 250 | 30 | NI | [53] |
| Taiwan | 300 | 5 | NI | NI | 600 | 60 | NI | [53] |
| Nigeria | 85 | 3 | NI | 100 | 300 | NI | 100 | [54] |
| German | 50 | 1 | NI | NI | 150 | 70 | NI | [53] |
| Finland | 750 | 20 | NI | 200 | 400 | 100 | 300 | [55] |
| Netherland | 50 | 1 | NI | 50 | 200 | 20 | 100 | [56] |

\* General resource information standard guideline can be found via the link https://esdat.net/environmental-standards/.

NI = not indicated.

**Table 2. Maximum standard permissible heavy metal concentrations (mg L$^{-1}$) in water.**

| Organization | Pb | Cd | Mn | Cu | Zn | Cr | Reference |
|---|---|---|---|---|---|---|---|
| US EPA | 0.05 | 0.005 | NI | NI | NI | 0.1 | [49] |
| WHO | NI | 0.005 | NI | 2 | 5 | 0.05 | [57,58] |
| FAO/WHO | NI | | 0.2 | 75 | | NI | [51] |
| TBS | 0.05 | NI | NI | NI | NI | NI | [50] |

NI = not indicated.

 

Bits), and the Fischer's Least Significant difference (L.S.D) was used to compare treatment means at p = 0.05 level. The mean concentrations of elemental compositions were then compared with TBS standard maximum tolerable limits using one-sample t-test at 95% confidence level [44]. To further understand inter-element relationships, a correlation coefficient was employed to evaluate associations between pairs of heavy metals and between heavy metals and metal oxides. The PCA was used in the analysis as it reduces data dimensionality and to identify patterns in metal distributions, potentially indicating shared sources or similar behaviors [60,61].

## Results and discussion

### Analysis of heavy metals in the soil

The average levels of heavy metals in the soil (mg kg$^{-1}$) from different sampling locations in Bahi villages showed varying concentrations (Table 3). The locations BMS and BSS showed higher mean concentrations of Cd. The compositions of heavy metals analyzed in soil samples were Pb, Cd, Mn, Zr, Cu, As, Zn, Sr and Cs; and in water samples were Mn, Zn and Cu. The levels of heavy metals in soil from sampled locations in Bahi district varied and mean concentrations increased in the order trend of As < Pb < Zn < Cu < Sr < Cs < Zr < Cd < Mn (Fig 2 and 3 and S1-S5 Figs). The obtained concentrations of different heavy metals were recorded against geological coordinates for each sampling point presented in Table C (S1 Table) The results from the analyzed metal oxides are also discussed. The map of concentration range (mg kg$^{-1}$) of analyzed heavy metals for different sampling points are reported Figs A to H (S1 Fig). The observed higher standard deviation from the measured samples indicates a wider range of values. However, the mean concentrations were not above the permissible limit by different regulatory authorities; FAO/WHO, EU, US EPA, and TBS except for Cd as presented in Table 1, [44,62,63].

Lead (Pb) mean concentrations in the soil samples ranged from 39.00 ± 1.00 to 22.00 ± 2.65 mg kg$^{-1}$, the highest being from Bahi Matajila (BTS) and the lowest from Bahi Chang'ombe (BCS) observed with an overall mean concentration of 29.6 mg kg$^{-1}$. The analysis shows the mean concentrations of Pb in different locations are significantly different (one sample t-test, p < 0.05) at 95% confidence level. Despite the variations, the concentrations are within permissible limits set by environmental guidelines, such as the WHO/FAO standards of 100 mg kg$^{-1}$ for agricultural soils [48]. Factors influencing the lead speciation include soil pH, organic matter, presence of various amendments, clay minerals and presence of organic colloids and iron oxides [64,65]. The variability could be influenced by the natural geochemical processes. Commonly, Pb is known to have low mobility in soil, hence its presence at elevated levels could suggest point-source contamination, potentially from agricultural inputs or industrial emissions. Pb poses serious health risk to both adults and children

**Table 3. Concentrations of heavy metal in soil sample (Mean±SD mg kg$^{-1}$).**

| Locations | Pb | Cd | Mn | Zr | Cu | As | Zn | Sr | Cs |
|---|---|---|---|---|---|---|---|---|---|
| **BNS** | 34.00 ± 13.00 | 51.33 ± 9.24 | 521.33 ± 66.01 | 215.33 ± 15.53 | 47.33 ± 7.51 | 3.67 ± 2.87 | 46.00 ± 11.36 | 152.33 ± 6.03 | 132.33 ± 8.33 |
| **BCS** | 22.00 ± 3.00 | 65.67 ± 1.15 | 271.33 ± 39.68 | 98.33 ± 61.72 | 58.33 ± 6.11 | 4.33 ± 1.15 | 21.67 ± 5.51 | 77.00 ± 4.58 | 148.33 ± 11.93 |
| **BTS** | 39.00 ± 1.00 | 38.00 ± 1.00 | 882.00 ± 1.00 | 161.00 ± 1.00 | 57.00 ± 1.00 | 8.00 ± 1.00 | 101.00 ± 1.00 | 135.00 ± 1.00 | 121.00 ± 1.00 |
| **BMS** | 22.00 ± 2.65 | 704.00 ± 7.09 | 306.67 ± 77.49 | 252.33 ± 41.88 | 49.33 ± 6.67 | 5.33 ± 2.08 | 28.67 ± 5.86 | 88.33 ± 7.64 | 141.33 ± 11.02 |
| **BSS** | 31.00 ± 1.00 | 1157 ± 1.00 | 410.00 ± 1.00 | 303.00 ± 1.00 | 48.00 ± 1.00 | 5.00 ± 1.00 | 45.00 ± 1.00 | 157.00 ± 1.00 | 127.00 ± 1.00 |
| *Mean* | 29.60 | 403.20 | 478.27 | 206.00 | 52.00 | 5.26667 | 48.46667 | 121.93333 | 134.00 |
| *F Value* | 4.51604 | 43068.05 | 75.95 | 16.31221 | 2.94643 | 2.57292 | 74.51704 | 174.40793 | 5.37313 |
| **P** Value | 0.02423 | 3.95E-21 | 1.92E-07 | 2.21E-04 | 0.0755 | 0.10278 | 2.10E-07 | 3.34E-09 | 0.01424 |

Analysis of heavy metals in soil sample: Bahi Nagulo soil sample (BNS); Bahi Chang'ombe soil sample (BCS); Bahi Matajila soil sample (BTS); Bahi Makulu soil sample (BMS); Bahi Sokoni soil sample (BSS). value reported as mean ± SD; n = 3. Mean followed by dissimilar letter in a row are significantly different from each other at p < 0.05 according to Fischer Least Significant Difference (LSD).

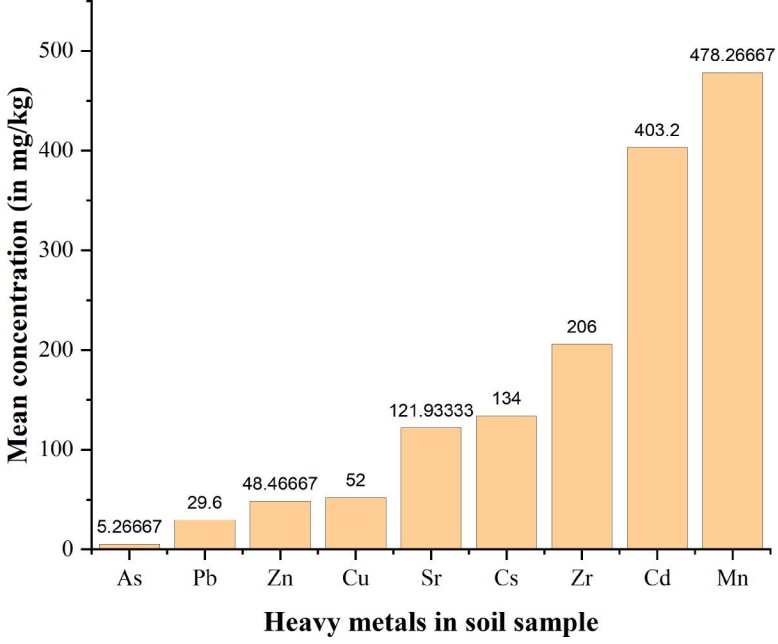

**Fig 2. The increasing order of mean concentrations of heavy metals in soil from different locations of Bahi.**

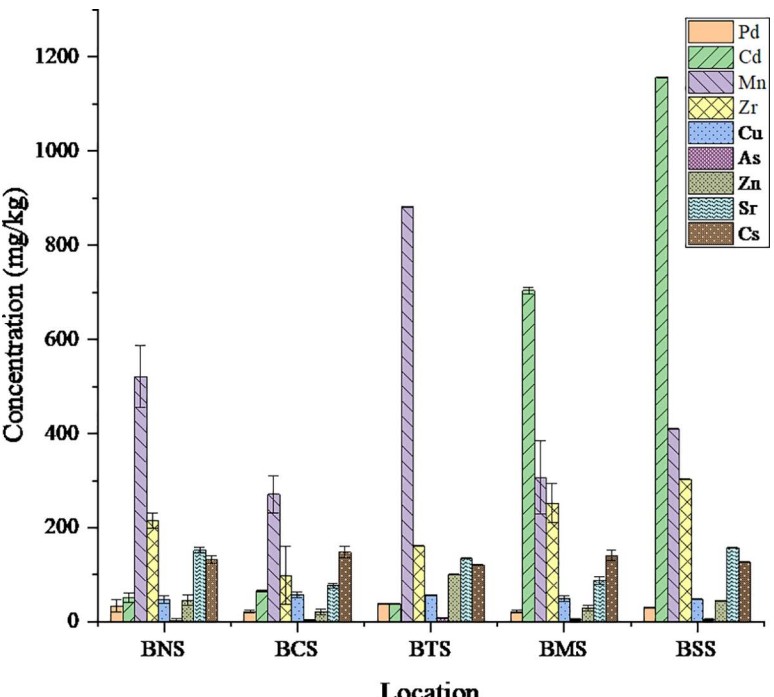

**Fig 3. Mean concentrations of heavy metals in soil from different locations of Bahi.**

when is at high concentrations; it is associated with impaired fertility and complicated pregnancy, cardiovascular diseases, impaired kidney function, high blood pressure and neurodevelopmental effects [66].

The average levels of Cd in Bahi Sokoni (BSS) and Bahi Matajila (BTS) samples showed high concentration of $1157.00 \pm 1.00$ mg kg$^{-1}$ and $704.00 \pm 7.09$ mg kg$^{-1}$, respectively, which was more than 20 times higher than the samples BNS, BCS, and BMS from the three locations Bahi Nagulo, Bahi Chang'ombe and Bahi Makulu, respectively. The statistical analysis shows Cd had minimal p value ($p < 0.05$, F = 43068.05), indicating the highest variability, thus a significant difference in various locations. The mean average of the Cd was found to be $403.20 \pm 507.44$ mg kg$^{-1}$. However, statistically, the mean concentration of Cd at Bahi district does not differ significantly with the maximum tolerable limit set by TBS (Wilcoxon rank sum test, $p > 0.05$) at 95% confidence level. The observed higher standard deviation from the measured samples indicates a wider range of values, that, the distribution of Cd in the respective areas was heterogeneous. The spatial distribution of Cd concentrations and the spatial distribution of Cd accumulations have been reported in the provinces in northern China having lower Cd concentrations than those in southern China ranging from 0.06 to 4.17 mg kg$^{-1}$ [67]. Usually, Cd concentrations in the soil decrease with increasing soil depth which is dependent upon parent materials. Normally, in natural soil where there is no anthropogenic activity that causes pollution, Cd concentration ranges from 0.01 to 2.7 mg kg$^{-1}$ [68,69]. Among several factors which affect Cd concentration in soil, parent rock material, high elevation, temperate, and dryer climate are profoundly considerable [24,69]. Cadmium has a potential to accumulate in tissues of aquatic organisms which can in turn lead to bio magnification, It has a biological half-life of about 30 years in humans, long term exposure can cause kidney damage and also painful in bones and in joints (Hodgson, 2010). In general, cadmium can be present in soil due to natural processes or contamination from sources like industrial emissions and agricultural practices [70]. Its exposure can lead to kidney damage, lung cancer, and other health complications [71].

Manganese (Mn) concentration in the soil samples ranged from $882.00 \pm 0.1$ to $271.33 \pm 39.68$ mg kg$^{-1}$, Bahi Matajila (BTS) having the highest concentration and Bahi Chang'ombe (BCS) with the least concentration. Compared to all other analyzed heavy metals, Mn had the highest mean concentration. However, all the concentrations are within the permissible limit of $10 \times 10^4$ to $1.5 \times 10^3$ mg kg$^{-1}$ according to TBS (TZS 972:2020(E), and of $1.5 \times 10^3$ mg kg$^{-1}$, $2.0 \times 10^3$ mg kg$^{-1}$ according to EU, WHO, respectively [50,51]. The mean average of $478.27 \pm 245.86$ mg kg$^{-1}$ with $p < 0.05$ shows large variation of concentrations within the sampled location and significantly different. Despite the fact that Mn is the major element, its ores (pyrolusite), manganese oxides, are known to be stable in both acidic and alkaline medium hence account for poor solubility and readily available in soil [24,72].

Zirconium (Zr) is another trace element observed in the soil sample at highest concentration was $301.00 \pm 5.25$ mg kg$^{-1}$ and the lowest $98.33 \pm 61.71$ mg kg$^{-1}$ for Bahi Sokoni and Bahi Chang'ombe, respectively. This range was within the recommended tolerable limit of 32–850 mg kg$^{-1}$ in soil [73]. Zr primarily exists as Zr$^{4+}$ in minerals, forming stable silicates like zircon (ZrSiO$_4$), which dominate Zr content in igneous rocks. It concentrates in late-stage magmatic rocks (e.g., granites) and is highly resistant to weathering, resulting in its enrichment in resistate-rich sediments [74]. The distribution of Zr in different sampled locations indicated to be significantly different ($p < 0.05$). Despite its low mobility and phytoavailability in soils, Zr can be absorbed by plants, mainly accumulating in root cells, and while generally low in phytotoxicity, it can still reduce plant growth and affect enzyme activity, necessitating further studies on its environmental behavior and impact [73]. Likewise, Zr is known to be retained in significant amounts in biological systems, but no known metabolic function [73].

Zinc (Zn) concentrations in the soil samples varied significantly across the different locations. The mean concentrations ranged from $21.67 \pm 5.51$ mg kg$^{-1}$ in Bahi Chang'ombe (BCS) to $101.00 \pm 1.00$ mg kg$^{-1}$ in Bahi Matajila (BTS), with an overall mean concentration of 48.47 mg kg$^{-1}$. The analysis indicates significant differences in Zn concentrations among the locations with a p-value of $2.10 \times 10^{-7}$ ($p < 0.05$). The maximum mean concentration is Zn is within permissible limits by regulatory bodies such as the FAO/WHO and TBS standards, 300 mg kg$^{-1}$ and 150 mg kg$^{-1}$, respectively. The observed variability suggests a combination of natural influences of geochemical ecology including soil pH, and organic matter

content. In areas like Bahi Chang'ombe (BCS) with low concentration of zinc could reflect natural geochemical variability since its mobility in soil depends largely on its interaction with organic colloids and clay minerals, and its availability to plants is closely tied to soil pH [7]. Despite of the importance of zinc to plants and animals as essential micronutrient for plants and animals it can become toxic at elevated levels. Although Zn is less harmful than some heavy metals, elevated concentrations can pose ecological and health risks, including phytotoxicity and potential bioaccumulation in the food chain [75]. However, since the mean concentrations are within the permissible limits, no danger has been predicted so far that could require mitigation intervention for risks associated with high zinc concentrations in the area.

Arsenic (As) was among the least of the analyzed elements, its concentration ranged from $8.00 \pm 1.00$ to $3.67 \pm 2.87$ mg kg$^{-1}$. With regard to drinking water, arsenic is one of the heavy metals of great concern since it is associated with bladder cancer [76]. A study by Knivsland (2012), reported arsenic in water samples from wells with high concentration range from 2.7 to 0.3 µg L$^{-1}$ at Chipanga B, Bahi wetlands with explanation that it might have been contributed by poorly sorbed on iron hydroxides like goethite [77]. In other parts of the world where arsenic is observed at higher concentrations than permissible limits were chiefly associated with enrichment in the topsoil [78]. In the case of Bahi localities, no evidence uses anthropogenic activities or topsoil enrich. Arsenic is considered to be a "class A" human carcinogen element and its tolerance concentration limits in topsoil vary from 0.8–20 mg kg$^{-1}$ for some states in the USA to 25–50 mg kg$^{-1}$ in Canada [79]. Chronic exposure to arsenic can lead to different cancers such as skin, lungs and lymph glands [76]. On the other hand, acute toxicity is associated with vomiting, watery and bloody diarrhoea, severe abdominal pain, and burning oesophageal pain [80]. However, the findings from this study revealed that the concentration of As were within the tolerable limits.

Copper (Cu) observed at concentrations ranged from $58.33 \pm 6.11$ to $47.33 \pm 7.51$ mg kg$^{-1}$ which it is within permissible limit of 200 mg kg$^{-1}$ but according to international standards it is above the limit by at least 56% [50,51]. Despite the fact that, trace elements such as Cu is essential for microorganisms due to its vital role in cofactors for metallo-proteins and enzymes, above the recommended thresholds concentrations they become toxic to both humans and non-human animals [13,14].

Strontium (Sr) were also observed at mean concentration maximum $157.00 \pm 1.00$ to 77 mg kg$^{-1}$. Despite variations of Sr in soils, natural strontium belongs to a microelement and comprises a mixture of four stable isotopes: $^{84}$Sr (0.56%), $^{86}$Sr (9.96%), $^{87}$Sr (7.02%) and $^{88}$Sr (82.0%)., the typical concentration is 0.2 mg kg$^{-1}$ [81]. Research indicates that the concentration of $^{90}$Sr in soil is minimal because a significant fraction of both stable and radioactive strontium dissolves in water [82,83]. Consequently, this mobility facilitates its migration into deeper soil layers, increasing the likelihood of its entry into groundwater system. The mean concentration of Sr from the studied area still with the tolerable limit since Sr appears on average of 370 mg kg$^{-1}$ in rocks and 240 mg kg$^{-1}$ in soils [83], and in comparison other studies show Sr levels in surface horizons vary from 715–1000 mg kg$^{-1}$, in America ranges from 110–445 mg kg$^{-1}$ [84]. The Sr concentration in plants is highly variable the calculated mean for different food and feed plants, range from about 10–1500 mg kg$^{-1}$ dry weight whereby the lowest mean contents of Sr were found in fruits, grains, and potato tubers, whereas legume herbage contained from 219 to 662 mg kg$^{-1}$. The concentrations of Sr in blueberries have been reported ranging from 4.5–5.5 mg kg$^{-1}$ and 2.9–3.9 mg kg$^{-1}$ for Russia (near Moscow) and Germany, respectively [84]. Sr is potentially harmful to plants; Its uptake by roots is apparently related to both the mechanisms of mass-flow and exchange diffusion, thus, it mobility and availability to plant roots in soil are controlled by external factors such as chemical composition of the soil and pH, temperature and agricultural soil cultivation as well as soil biological networks built by microbial communities [85,86].

Cesium (Cs) was found to have mean concentration ranging from $148.33 \pm 11.93$ to $121.00 \pm 1.00$ mg kg$^{-1}$, although there is no strict threshold for cesium in soil, but the presence of radioactive cesium ($^{137}$Cs) above background levels may indicate contamination. Geochemically, it shows a stronger tendency to bind with aluminosilicates, predominantly concentrated in acidic igneous rocks and argillaceous sediments, alongside other monovalent trace cations of Rb, K, Na and Li. However, it is not very chemically reactive and has resemblance with potassium in its chemistry in soil. Cs is strongly adsorbed in soil and is released through weathering though information on its distribution in soil remain limited

[87]. Literatures show that Cs concentrations in soils range widely, the value range between 0.3 and 26 mg kg⁻¹ [88]; the range compares favorably with the levels reported for Canadian reference soils, Bulgarian soils, and Germany & Sweden range from 0.3 to 5.1 mg kg⁻¹, 2.2 to 16.7 mg kg⁻¹, and 0.1 to 1 mg kg⁻¹, respectively [84]. However, the values observed from this study were 5 times higher, that may be due to geological characteristics of the underlaying rocks of Bahi district, and organic-rich soil horizons, potential factors for Cs distribution and accumulation [89]. Also in correlation analysis, the content of clay minerals also showed a significant positive correlation with $^{137}$Cs activity concentration [90].

The distribution levels of heavy metals whether influenced by anthropogenic activities or natural background sources, have generally remained below the maximum permissible limits established by both national and international regulatory authorities. The analysis shows the mean concentrations of Pb, Mn, Cu, Zn and As at Bahi district differ significantly with the maximum tolerable limits set by TBS (one sample t-test, $p < 0.05$) at 95% confidence level. The following is a summary of the mean concentrations reported in other studies across Tanzania).

## Analysis of metal oxides concentrations in the soil

The mean concentrations of metal oxides ($SiO_2$, $Fe_2O_3$, $Al_2O_3$, $CaO$, $Ti_2O$, $KO_2$ and $BaO$ and $TeO_2$) were determined. Most oxides in soil occur as secondary minerals with exception of silicon oxides and titanium oxides which are predominantly inherited from the parent rock. Table 4 and Fig 4 summarizes the mean concentrations (mg kg⁻¹) of metal oxides determined in soil samples from different locations; and Fig 5 for $SiO_2$ and $TeO_2$ which the later was commonly highly distributed in all samples.

The analysis of soil samples revealed varying mean concentrations, $SiO_2$ from all locations were 90.11 ± 1.97 mg kg⁻¹. The highest mean concentration of $SiO_2$ in soil was 91.81 ± 1.92 mg kg⁻¹ for BCS samples and the lowest was 87.06 ± 0.087 mg kg⁻¹ from BTS. Silicon is the second most abundant element (~28%) in the Earth's crust after oxygen, and is mainly found in the silica ($SiO_2$) or silicate ($SiO_4$) forms. It is a major constituent of nearly all rocks, sedimentary rocks, and felsic and intermediate igneous rocks [91]. The study conducted in New Jersey for agricultural importance of silicon revealed that the concentration of elemental Si is dependent upon the availability of their oxides; acetic acid extract exhibited ranges of soil test silicon from 4 to 35 mg L⁻¹, with the average soil test silicon level being 14 mg L⁻¹ [92]. It's toxicity in the soil depends on various factors including particle size, concentration, and exposure duration with which 0.1–10% accumulated Si in plant tissue have observed to be adequate to growth [93,94].

Tellurium oxides observed with the mean concentration in soil from all sampled locations was 254.4 ± 21.90 mg kg⁻¹ the highest was 2.72.67 ± 9.01 mg kg⁻¹ and the lowest 218.0 ± 1.0 mg kg⁻¹ for BCS and BTS sample, respectively. Tellurium,

**Table 4. Mean concentrations of metal oxides (mg kg⁻¹) of different locations.**

| LOCATION | SiO₂ | Fe₂O₃ | Al₂O₃ | CaO | K₂O | TiO₂ | BaO | TeO₂ |
|---|---|---|---|---|---|---|---|---|
| BNS | 89.55 | 3.30 | 1.21 | 0.33 | 0.71 | 0.15 | 0.13 | 252.00 |
| BCS | 91.81 | 1.44 | 0.40 | 0.21 | 0.57 | 0.05 | 0.11 | 272.67 |
| BTS | 91.81 | 8.46 | 1.88 | 0.25 | 0.57 | 0.32 | 0.12 | 218.00 |
| BMS | 90.34 | 2.32 | 1.59 | 0.60 | 0.75 | 0.07 | 0.10 | 260.00 |
| BSS | 87.06 | 2.35 | 3.17 | 0.49 | 1.37 | 0.03 | 0.12 | 269.33 |
| *Mean* | 90.12 | 3.57 | 1.65 | 0.38 | 0.79 | 0.13 | 0.12 | 254.40 |
| *SD* | 1.97 | 2.81 | 1.02 | 0.16 | 0.33 | 0.12 | 0.01 | 21.90 |
| *Sum* | 450.58 | 17.86 | 8.25 | 1.88 | 3.97 | 0.63 | 0.59 | 1272.00 |
| *Min* | 87.06 | 1.44 | 0.40 | 0.21 | 0.57 | 0.03 | 0.10 | 218.00 |
| *Max* | 91.81 | 8.46 | 3.17 | 0.60 | 1.37 | 0.32 | 0.13 | 272.67 |

Analysis of metal oxides in soil samples from; Bahi Nagulo soil sample (BNS), Bahi Chang'ombe soil sample (BCS), Bahi Matajila soil sample (BTS), Bahi Makulu soil sample (BMS), Bahi Sokoni soil sample (BSS); value reported as mean ± SD; n = 3.

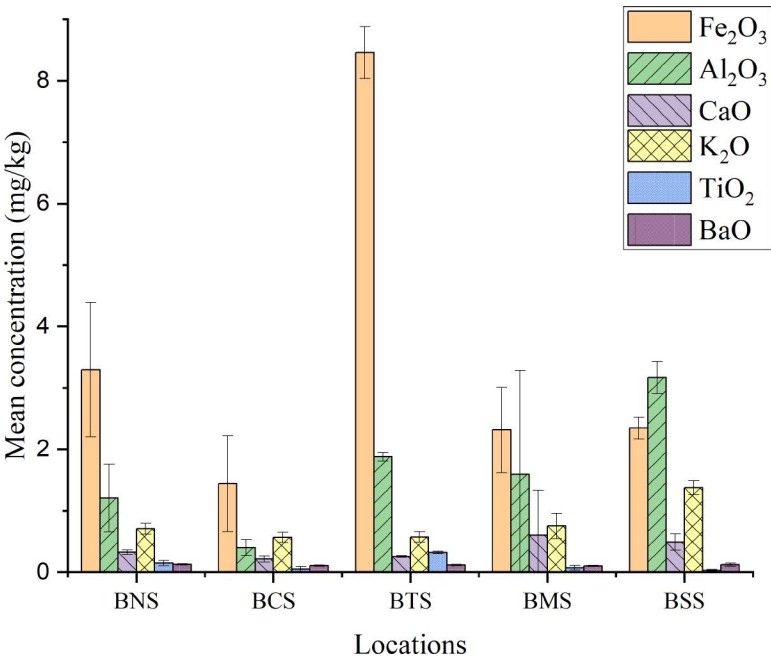

**Fig 4. Mean concentrations (mg kg$^{-1}$) of metal oxides in soil samples from different locations.**

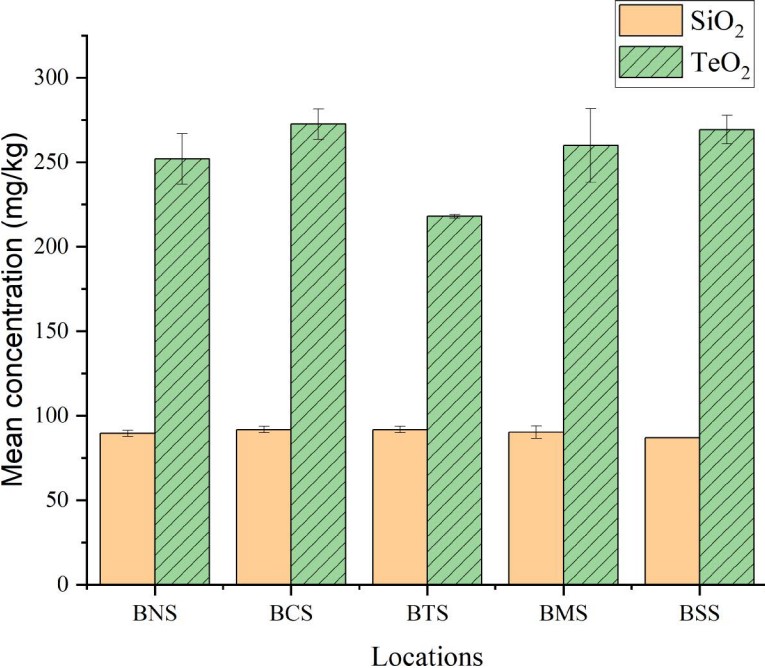

**Fig 5. Mean concentrations (mg kg$^{-1}$) of SiO$_2$ and TeO$_2$ in soil samples from different locations.**

typically present in soil at very low concentrations as Te(IV) and Te(VI), it is both toxic and teratogenic, with evidence suggesting that Te(IV) may be more toxic than Te(VI) [95,96]. Its abundance ranges from 0.001 to 0.005 mg kg$^{-1}$; despite its low in abundance, tellurium and its compounds can be toxic to humans when ingested, inhaled, or absorbed through the skin [96,97]. The mean concentration of Te observed from this study is quite high compared to most reported values in literature for naturally occurrence in Earth's crust [98,99].

The mean concentration of $Fe_2O_3$ in soil sample from all locations was 3.57 ± 2.81 mg kg$^{-1}$, the highest was 8.46 ± 0.42 mg kg$^{-1}$ and the lowest was 1.44 ± 0.78 mg kg$^{-1}$ for BTS and BSS samples, respectively. Studies show that oxides of iron (II) in soil generally exhibit a high specific surface area with relative surface sites which bind strongly with oxy-anions (such as $AsO_4^{3-}$) and metal cations such as $Pb^{2+}$, $Zn^{2+}$, $Cd^{2+}$, and $Cu^{2+}$, a tendency which affects mobility of plant nutrients and toxic metals [100]. Usually iron (III) oxide favor the binding of arsenic, but the reducing condition, high pH and anaerobe conditions may favor its dissociation [77,101–103]. However, increase of an oxidizing condition in soil can dissolve Fe-Mn bond and organic matter fractions, thus presence of metal oxides contribute much to the reduction of heavy metal up-take by plants [104].

The mean concentration of CaO in soil for all sampled locations was 0.38 ± 0.16 mg kg$^{-1}$, the highest value was 0.60 ± 0.74 mg kg$^{-1}$ and the lowest 0.21 ± 0.05 mg kg$^{-1}$ for BMS and BCS samples, respectively. Different rocks are responsible for availability of CaO such as gneisses, apatite, carbobatite which are also found in Bahi district [24,105]. In soil ecology, CaO can reduce heavy metals' stress to plants, for example, the study involving CaO showed that the root treatment of 20 mg L$^{-1}$ calcium oxide nanoparticles (CaONPs) and 2% farmyard manure reduced the cadmium acquisition from the soil and improved growth in height by 27.4% compared to positive control under Cd stress [104,106]. Other studies show that CaO application is effective in mitigating Cd transfer from paddy soil to rice plant and $CaCl_2$ extraction gives a reliable prediction of soil Cd bioavailability to paddy rice [107]. A research conducted on the role of CaO as a curing agent contaminated soil, when added to $Zn^{2+}$-contaminated red clay, CaO will react with $SiO_2$, $Al_2O_3$, and $Fe_2O_3$ in the red clay to produce calcium aluminate hydrate (C-A-H) and calcium silicate hydrate (C-S-H), which has a better effect on the strength of the soil, better for the low concentration of contaminated soil and weaker for the high concentration of contaminated soil [108]. Therefore, presence of naturally occurring CaO in the Bahi environment can be potentially needful as long as Bahi area is characterized by the presence of radionuclides and heavy metals.

Oxides of potassium ($K_2O$) was found to have a mean concentration of 0.79 ± 0.33 mg kg$^{-1}$ of which the highest value was 1.37 ± 0.27 mg kg$^{-1}$ and the lowest 0.57 ± 0.09 mg kg$^{-1}$ for BSS and BCS samples, respectively. $TiO_2$ observed with the mean concentration for all locations, the highest value was 0.13 ± 0.12 mg kg$^{-1}$ and the lowest was 0.03 ± 0.02 mg kg$^{-1}$ for BTS and BSS samples, respectively. Geochemically, most of the $TiO_2$ minerals in soil range from 50 to 125 μm in size. In the European soils, the median Ti concentration in topsoils and sediments is found to be around $6 \times 10^3$ mg kg$^{-1}$, however, as the technology advances the $TiO_2$ engineered becomes a challenge to distinguish from the natural particles, since they do not differ by composition, but on diverse morphologies [109]. However, these results from Bahi do not suggest any potential health risks as long as it is naturally occurring unlike the once that are exposed to human-related activities [110].

The mean concentration of BaO in soil samples from all locations was 0.12 ± 0.01 mg kg$^{-1}$ with which the highest mean concentration was 0.13 ± 0.01 mg kg$^{-1}$ and the lowest 0.10 ± 0.01 mg kg$^{-1}$ for BNS and BMS samples, respectively. BaO can react with heavy metals to form insoluble compounds, which reduces their mobility and bioavailability in soil, thus, preventing heavy metals from leaching into groundwater and reduce plant uptake [111].

## Analysis of heavy metal concentrations in water

The water samples were analyzed for Mn, Zn and Cu. Mn showed high concentration compared to Zn (Table 5 and Fig 6) whereas Cu concentrations were below detectable limit (BDL) of 0.01 mg L$^{-1}$. All sampling geographical coordinates for each sample were recorded against the analyzed heavy metal concentrations as presented in Table G. (S1 Table)

The analysis of Mn from water samples showed the maximum mean value concentration of 0.64 ± 0.01 mg L$^{-1}$ and the minimum value of 0.05 ± 0.02 mg L$^{-1}$, for samples W-01 and W-14, respectively. Zn was detected in water from sites

**Table 5. Mean concentrations (mg L⁻¹) of Mn and Zn in water samples.**

| Sample | Mn | Zn |
|---|---|---|
| W-01 | 0.64±0.01 | 0.01±0.00 |
| W-03 | 0.53±0.03 | 0.01±0.00 |
| W-05 | 0.22±0.01 | 0.01±0.00 |
| W-07 | 0.37±0.01 | 0.17±0.05 |
| W-09 | 0.27±0.02 | 0.12±0.01 |
| W-13 | 0.26±0.02 | 0.19±0.01 |
| W-14 | 0.05±0.02 | 0.13±0.01 |
| *Mean* | 0.33 | 0.09 |
| *SD* | 0.2 | 0.08 |
| *Sum* | 2.34 | 0.64 |
| *Min* | 0.05 | 0.01 |
| *Max* | 0.64 | 0.19 |

Water samples; River Bubu (W-01), Bahi Swamp (W-03), Bahi town (W-05), Chali town (W-07); Bahi Makulu (W-09), Bahi Sokoni (W-013) and Bahi sokoni (W-014) Value reported as mean±SD; n=3. BDL = Below detectable limit.

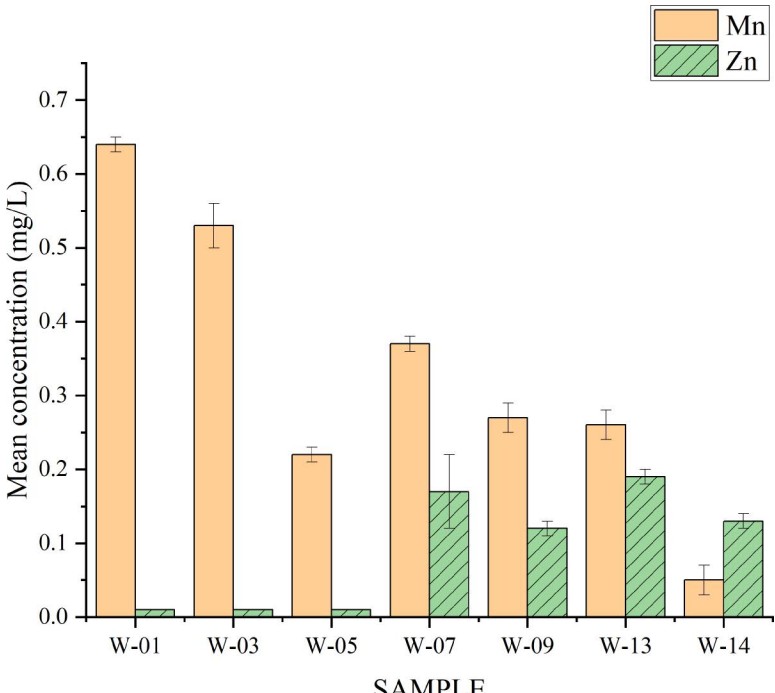

**Fig 6. The mean concentrations of Mn and Zn in water from different water samples.**

W-07, 09, 13 and 14 with mean concentrations range from 0.13±0.01 to 0.19±0.01 mg L⁻¹. The concentrations in other locations, W-01, W-02 and W-03 were below the detectable limit of 0.01 mg L⁻¹. Thus, the mean concentrations of Mn and Zn in water samples were 0.33±0.20 and 0.09±0.08 mg L⁻¹, respectively. Since the computed independent t-test (3.003, p=0.011) shows there is a statistically significant difference between the mean for mean concentrations of 0.33

and 0.09 mg/L for Mn and Zn, respectively. However, the results do not suggest any potential risk because the observed concentrations were below the permissible limits. As explained earlier, assessment of Mn levels in water is crucial since despite of importance of Mn in enzymatic reactions, its accumulation behaviour in endothelial cell can in turn lead to damage of their membranes, resulting into haemorrhage and edema [76]. Furthermore, Mn can cause diseases associated with nerve system [112].

## Correlation analysis

Determination of the association between elements which occur naturally in the environment is of paramount importance, since their concentrations spell out their common origin. Pearson Correlation Coefficient) was determined to visualize correlations among heavy metals and among common metal oxides as presented in Table 6. These correlations were also observed from the PCA plots, by using the Kaiser Criterion, eigenvalues were greater than 1, hence considered significant suggesting that PC1 and PC2 capture a substantial amount of the variance, justifying their inclusion in visualization. (Figs 7–9).

The analysis of the correlation coefficient, r, was used to determine the associations between heavy metals, and with metal oxides. Studies show that high correlation coefficient (r = near +1 or −1) means a good relation between two variables, and its concentration around zero means no association between them at a significant level of 0.05. In these results, it has been considered that, heavy metals are strongly correlated (if r = above 0.6, $p < 0.05$), moderately correlated (when r = positive but less than 0.6, $p < 0.05$), and weakly correlated (r = negative value). For correlation between metal and metal oxides (if r = above 0.4, $p < 0.05$), moderate correlated (when r = positive but less than 0.4, $p < 0.05$), and weak correlation (r = negative value). For correlation between metal oxides, (if r = above 0.7, $p < 0.05$), moderate correlated (when r = positive but less than 0.7, $p < 0.05$), and weak correlation (r = negative value). Principal component analysis (PCA) plots (Figs 6–8) were also used to visualize the relationships between different heavy metals, metal oxides and between metals with metal oxides. The Kaiser–Meyer–Olkin Measure for Sampling Adequacy (KMO) and Bartlett's test of data sphericity was performed to evaluate the validity of PCA approach. The KMO value (0.577) > 0.5 and $p < 0.01$ indicated the need to undertake PCA.

**Correlation between metals.** Pearson correlation analysis, revealed positive correlation between concentrations of heavy metals. Results which show strong positive correlation were observed between Mn with Zn (r = 0.987), Pb with Mn (r = 0.917), Pb with Zn (r = 0.889), Cd with Zr (r = 0.847), As with Zn (r = 0.833), Pb with Sr (r = 0.813), and Mn with As (r = 0758) (Fig 7). This strong positive correlation shows that the elements are closely associated, thus suggesting their common origin. However, other studies of soil profiles have shown Mn having a moderate correlation with metals Zr correlation coefficient range between 0.4 and 0.6, $p < 0.05$ significant level [113,114]. Nevertheless, the geochemical nature of Bahi wetland supports the association of these metals due to the presence of carbobatites, zircon, and granites [24,115,116]. With the exception of Zr, Cd showed either weak or negative correlations with the other trace metals studied, consistent with findings reported in similar research [117,118].

**Metals with metal oxides correlation.** Strong positive correlation was observed for most of the analyzed metal oxides with metals, thus, the very strongly correlation $Fe_2O_3$ with Zn (r = 0.981), $Fe_2O_3$ with Mn (0.978), $TiO_2$ with Mn (r = 0.953), $TiO_2$ with Zn (r = 0.922), BaO with Sr (r = 0.907), $K_2O$ with Cd (r = 0.905), and $SiO_2$ with Cu (r = 0.812) (Fig 8). Studies show negative correlation between $SiO_2$ with Sr (r = − 0.40) and a moderate positive correlation with Zr (r = 0.44) pointing to the mineralogical association of zircon with quartz [119]. Contrary to the findings, geochemical and biochemical characteristics of Sr are similar to those of Ca; thus, Sr is very often associated with Ca in the terrestrial environment. The weak correlation between Sr and CaO observed from this study it may be due to the reason that at an elevated CaO level tends to enhance Cd solubility upon significant acidification of $pH \geq 6.5$ [107]. The Sr to Ca ratio seems to be relatively stable in the biosphere and therefore is commonly used for the identification of built-up concentrations of Sr in a particular environment [84]. The analysis shows Sr to have a strong positive association with BaO; of which both Sr and Ba are

**Table 6. Correlation coefficient (r) matrix for heavy metal and with metal oxides concentrations in soil samples.**

|  | SiO$_2$ | Fe$_2$O$_3$ | Al$_2$O$_3$ | CaO | K$_2$O | TiO$_2$ | BaO | TeO$_2$ | Pb | Cd | Mn | Zr | Cu | As | Zn | Sr | Cs |
|---|---|---|---|---|---|---|---|---|---|---|---|---|---|---|---|---|---|
| SiO$_2$ | 1 | | | | | | | | | | | | | | | | |
| Fe$_2$O$_3$ | 0.3752 | 1 | | | | | | | | | | | | | | | |
| Al$_2$O$_3$ | −0.76801 | 0.1857 | 1 | | | | | | | | | | | | | | |
| CaO | −0.58537 | −0.35881 | 0.53791 | 1 | | | | | | | | | | | | | |
| K$_2$O | −0.94865 | −0.32759 | 0.85218 | 0.57369 | 1 | | | | | | | | | | | | |
| TiO$_2$ | 0.50579 | **0.97155** | −0.02438 | −0.46189 | −0.51026 | 1 | | | | | | | | | | | |
| BaO | −0.43742 | 0.29598 | 0.3254 | −0.31744 | 0.28265 | 0.29997 | 1 | | | | | | | | | | |
| Te | −0.42811 | −0.98106 | −0.08722 | 0.3187 | 0.42809 | −0.98754 | −0.26835 | 1 | | | | | | | | | |
| Pb | −0.09242 | 0.81531 | 0.40255 | −0.34138 | 0.03888 | 0.77991 | 0.78662 | −0.78072 | 1 | | | | | | | | |
| Cd | −0.80848 | −0.39795 | 0.78724 | **0.81408** | **0.90453** | −0.58686 | −0.12253 | 0.47048 | −0.23946 | 1 | | | | | | | |
| Mn | 0.24768 | **0.97762** | 0.23869 | −0.39899 | −0.23888 | 0.95314 | 0.49028 | −0.95454 | **0.91691** | −0.39245 | 1 | | | | | | |
| Zr | −0.88018 | −0.19024 | **0.82062** | **0.8461** | **0.82883** | −0.32664 | 0.21294 | 0.1949 | 0.09224 | **0.84662** | −0.12871 | 1 | | | | | |
| Cu | **0.81218** | 0.34716 | −0.49884 | −0.71863 | −0.62076 | 0.37685 | −0.35315 | −0.28248 | −0.04447 | −0.5768 | 0.24001 | −0.87707 | 1 | | | | |
| As | 0.36796 | 0.86095 | 0.31139 | −0.11067 | −0.18475 | **0.75648** | −0.12283 | −0.80141 | 0.50012 | −0.08957 | **0.75776** | −0.11436 | 0.46363 | 1 | | | |
| Zn | 0.20862 | **0.98069** | 0.33579 | −0.33289 | −0.15592 | **0.92239** | 0.42317 | −0.93671 | **0.88942** | −0.28194 | **0.98769** | −0.07621 | 0.26164 | **0.83329** | 1 | | |
| Sr | −0.63979 | 0.36206 | 0.67152 | 0.028 | 0.53396 | 0.28601 | **0.90797** | −0.31905 | 0.81299 | 0.21345 | 0.52933 | **0.54249** | −0.53073 | 0.07108 | 0.51148 | 1 | |
| Cs | 0.3048 | −0.76561 | −0.70165 | 0.0265 | −0.31449 | −0.65095 | −0.65499 | **0.70393** | −0.92463 | −0.1158 | −0.84164 | −0.37707 | 0.18278 | −0.60461 | −0.8666 | −0.83693 | 1 |

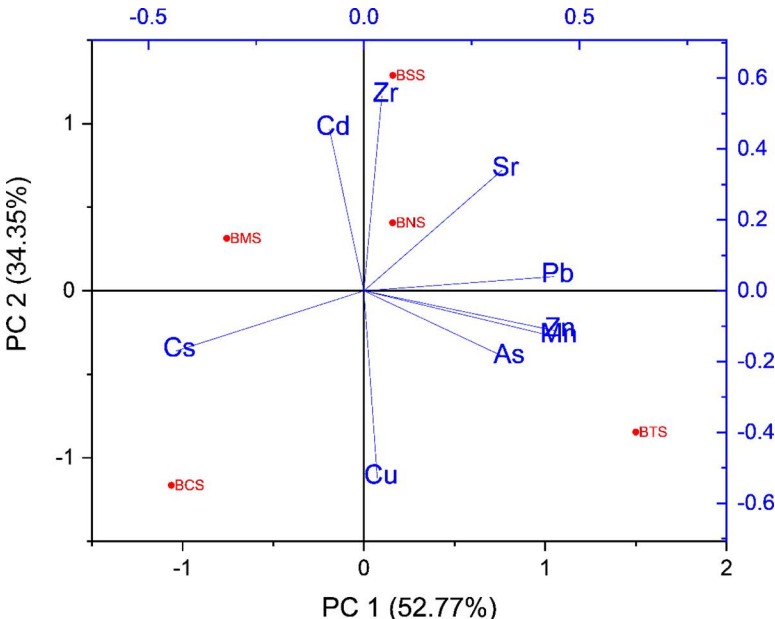

**Fig 7. Principal component analysis showing correlation between heavy metals.**

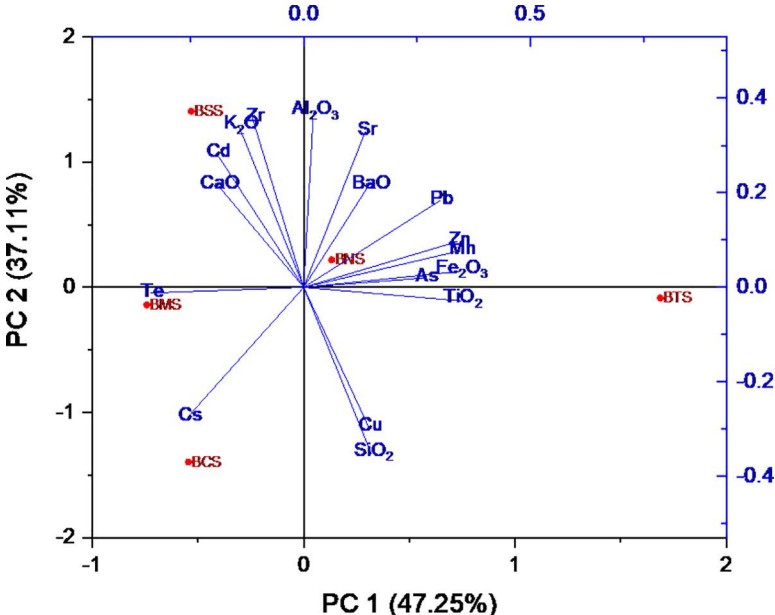

**Fig 8. Principal component analysis showing correlation between heavy metals and metal oxides.**

group II elements. Other studies show that there are strong correlations between metals (Cd, Cu, Mn, Pb, and As) with $Fe_2O_3$, this is suggested as in case favors the binding with As but the reducing condition and high pH may favor arsenic distribution [103,120].

**Metal oxides correlation.** According to the present study, the correlation between $TeO_2$ and $Fe_2O_3$ ($r = -0.98106$) contrary to Grygoyć (2021) who found a positive correlations; arguing that, Te is generally bound by sorption onto

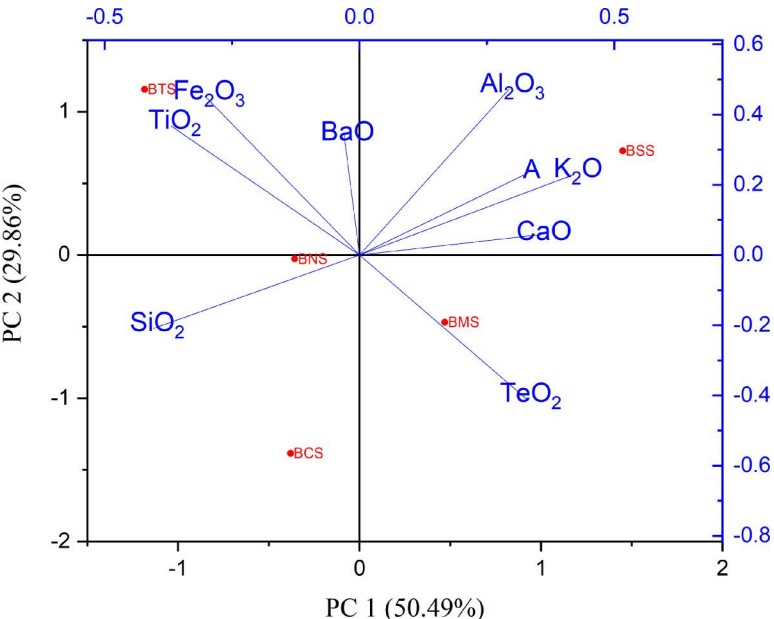

**Fig 9. Principal component analysis showing correlation between metal oxides.**

clay-sized soil particles rather than in minerals, which may imply Te(IV) is weakly bound to $Fe^{3+}$oxides by surface interactions only [46,99]. The correlation coefficients showed strong positive correlation between $Fe_2O_3$ with $TiO_2$ (r = 0.972), $Al_2O_3$ with $K_2O$ (r = 0.852), other showed moderate correlation and negative correlation (Table 7, Fig 9). Similarly, the correlation between metal oxides, were also reported for the research conducted at Manyoni, a district near Bahi wetland characterized with similar geochemical profiles [115].

## Conclusion

This study provides insights into the distribution and potential risks of heavy metals in the terrestrial environment, revealing geochemical variations of Bahi district influenced by both natural and anthropogenic factors. The exceptionally high

**Table 7. Comparison of mean concentration (mg kg⁻¹) in soil from Bahi district and other places reported in literature.**

| AREA | Heavy metals concentration (mg kg⁻¹) | | | | | | | | | Reference |
|---|---|---|---|---|---|---|---|---|---|---|
| | Pb | Cd | Mn | Zr | Cu | As | Zn | Sr | Cs | |
| **Bahi** | 29.60 | 403.20 | 478.27 | 206 | 52.00 | 5.27 | 48.47 | 121.93 | 134 | This study |
| *Mkuju* | 24.80 | 12.2 | 800.00 | NI | 8.70 | NI | 28.50 | NI | NI | [36] |
| *Morogoro* | 1.16 | 0.3 | NI | NI | 1.50 | NI | 24.80 | NI | NI | [121] |
| *Lupa Gold Field* | 80.00 | 0.10 | NI | NI | 200.00 | 0.70 | NI | NI | NI | [122] |
| *Lake Victoria Basin* | 65.6 | 0.55 | NI | NI | 26.10 | | 137.00 | NI | NI | [123] |
| *Lake Victoria Basin (paddy farms)* | 18.90 | 0.45 | NI | NI | 12.80 | | 59.80 | NI | NI | [124] |
| *Njombe* | 3.80 | | 361.80 | NI | 2.40 | NI | NI | NI | NI | [125] |
| *Geita* | 61.20 | NI | NI | NI | 66.20 | NI | 71.00 | NI | NI | [126] |
| **TBS** | 200 | 1 | 1.50E + 03 | | 200 | 1 | 150 | | | [50] |
| **FAO/WHO** | 100 | 3 | 1.00E + 05 | | 100 | NI | 300 | | | [51] |

NI = not indicated.

cadmium concentrations (28.7–31.2 mg kg$^{-1}$) in Bahi Sokoni and Bahi Matajila soils suggest probable health risks based on WHO risk models. Similarly, the elevated tellurium levels, surpassing natural background concentrations, trigger the need for further investigation due to its toxicological implications. While manganese remains within regulatory limits, its spatial variability suggests complex geochemical controls affecting metal stability and mobility, affecting geochemical ecology in both soil and waters. The observed distribution patterns indicate a strong link between local geological formations, influenced by uranium deposits, and possible anthropogenic contributions. The study establishes a baseline for evaluating potential health risks, aligning with internationally recognized standards; emphasizes the importance of integrating geochemical data into environmental monitoring frameworks and public health policies to mitigate exposure risks, ensuring long-term ecological and human health protection in Bahi District.

## Supporting information

**S1 Table. Supporting information Tables.**
(PDF)

**S1 Fig. Figures showing concentration ranges (mg kg$^{-1}$) of different elements in different sampling points.**
(PDF)

## Author contributions

**Conceptualization:** Dominic Parmena Sumary, Musa Chacha, Jofrey Raymond, Firmi Paul Banzi.

**Data curation:** Dominic Parmena Sumary.

**Formal analysis:** Dominic Parmena Sumary.

**Funding acquisition:** Dominic Parmena Sumary.

**Investigation:** Dominic Parmena Sumary.

**Methodology:** Dominic Parmena Sumary, Firmi Paul Banzi.

**Resources:** Dominic Parmena Sumary.

**Supervision:** Musa Chacha, Jofrey Raymond, Firmi Paul Banzi.

**Visualization:** Edwin Gomezulu.

**Writing – original draft:** Dominic Parmena Sumary.

**Writing – review & editing:** Jofrey Raymond, Firmi Paul Banzi.

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
