## [Decision Letter · Decision Letter 0]

7 Nov 2024

Dear Dr. Sumary,

Thank you for submitting your manuscript to PLOS ONE. After careful consideration, we feel that it has merit but does not fully meet PLOS ONE’s publication criteria as it currently stands. Therefore, we invite you to submit a revised version of the manuscript that addresses the points raised during the review process.

Below is a synthesis of the reviewers’ observations and recommendations, which I believe will enhance the manuscript’s clarity, methodological transparency, and impact on the field.

In summary, we recommend the following improvements:

Methodology: Elaborate on sampling specifics, quality control, and statistical methods.

Data Visualization: Use graphical aids to make data interpretation more intuitive.

Discussion Expansion: Provide a thorough exploration of health and environmental implications, drawing connections with prior research.

Language and Structure: Address grammatical and structural issues to meet journal standards, including reducing similarity and enhancing originality.

We look forward to receiving your revised manuscript.

Kind regards,

Elena Marrocchino

Academic Editor

PLOS ONE

Journal Requirements: When submitting your revision, we need you to address these additional requirements. 1. Please ensure that your manuscript meets PLOS ONE's style requirements, including those for file naming. The PLOS ONE style templates can be found athttps://journals.plos.org/plosone/s/file?id=ba62/PLOSOne_formatting_sample_title_authors_affiliations.pdf 2. In your Methods section, please provide additional information regarding the permits you obtained for the work. Please ensure you have included the full name of the authority that approved the field site access and, if no permits were required, a brief statement explaining why. 3. We note that the grant information you provided in the ‘Funding Information’ and ‘Financial Disclosure’ sections do not match.  When you resubmit, please ensure that you provide the correct grant numbers for the awards you received for your study in the ‘Funding Information’ section. 4. We note that your Data Availability Statement is currently as follows: All relevant data are within the manuscript and its Supporting Information files. Please confirm at this time whether or not your submission contains all raw data required to replicate the results of your study. Authors must share the “minimal data set” for their submission. PLOS defines the minimal data set to consist of the data required to replicate all study findings reported in the article, as well as related metadata and methods (https://journals.plos.org/plosone/s/data-availability#loc-minimal-data-set-definition).For example, authors should submit the following data: - The values behind the means, standard deviations and other measures reported;- The values used to build graphs;- The points extracted from images for analysis. Authors do not need to submit their entire data set if only a portion of the data was used in the reported study. If your submission does not contain these data, please either upload them as Supporting Information files or deposit them to a stable, public repository and provide us with the relevant URLs, DOIs, or accession numbers. For a list of recommended repositories, please see https://journals.plos.org/plosone/s/recommended-repositories.If there are ethical or legal restrictions on sharing a de-identified data set, please explain them in detail (e.g., data contain potentially sensitive information, data are owned by a third-party organization, etc.) and who has imposed them (e.g., an ethics committee). Please also provide contact information for a data access committee, ethics committee, or other institutional body to which data requests may be sent. If data are owned by a third party, please indicate how others may request data access.

**Additional Editor Comments:**

Dear Authors,

The Reviewers and I have considered your manuscript "Assessment of Heavy Metals in Soil and Water from Bahi District, Tanzania" (PONE-D-24-37705).

Below is a synthesis of the reviewers’ observations and recommendations, which I believe will enhance the manuscript’s clarity, methodological transparency, and impact on the field.

In summary, we recommend the following improvements:

Methodology: Elaborate on sampling specifics, quality control, and statistical methods.

Data Visualization: Use graphical aids to make data interpretation more intuitive.

Discussion Expansion: Provide a thorough exploration of health and environmental implications, drawing connections with prior research.

Language and Structure: Address grammatical and structural issues to meet journal standards, including reducing similarity and enhancing originality.

Reviewers' comments:

Reviewer's Responses to Questions

**Comments to the Author**

1. Is the manuscript technically sound, and do the data support the conclusions?

Reviewer #1: Partly

Reviewer #2: No

2. Has the statistical analysis been performed appropriately and rigorously?

Reviewer #1: No

Reviewer #2: Yes

3. Have the authors made all data underlying the findings in their manuscript fully available?

Reviewer #1: No

Reviewer #2: No

4. Is the manuscript presented in an intelligible fashion and written in standard English?

Reviewer #1: No

Reviewer #2: No

Reviewer #1: The manuscript deals with "Assessment of heavy metals in soil and water from Bahi district, Tanzania" the following comments and suggestions are need to address before its consideration.

1. The methodology section should provide more detailed descriptions of the sampling process, including the specific locations and conditions under which samples were collected. This will help readers understand the context and reliability of the data

2. While the manuscript mentions the use of descriptive statistics, ANOVA, and PCA, it would be beneficial to elaborate on the rationale for choosing these specific statistical methods. Additionally, including a brief explanation of how these analyses contribute to understanding the relationships between heavy metals and metal oxides would strengthen the analysis section

3. More emphasis on the quality control measures taken during the study, such as the use of blank solutions and recovery studies, would add credibility to the results. This could be integrated into the methodology section. https://link.springer.com/article/10.1007/s12011-024-04234-0,

https://www.sciencedirect.com/science/article/pii/S0278691524003727,

4. 1. What specific AAS techniques were utilized?, 2. How do findings compare to previous studies? 3. What quality control measures were implemented? Referred to below papers and cite them

https://www.sciencedirect.com/science/article/pii/S0946672X24001019,
https://www.sciencedirect.com/science/article/pii/S0946672X24000749,
https://www.sciencedirect.com/science/article/pii/S0889157523008177

5. The results section could be improved by including visual aids such as graphs or tables that summarize the heavy metal concentrations in both soil and water samples. This would make the data more accessible and easier to interpret for readers

6. The discussion should delve deeper into the implications of the findings, particularly regarding the potential health risks associated with heavy metal concentrations. It would be useful to compare the results with similar studies in other regions to provide a broader context

7. The manuscript mentions the presence of radionuclides and heavy metals in the Bahi area. Expanding on how these factors interact with local agricultural practices and their potential impact on the community would provide valuable insights

8. Including a section that outlines potential future research directions based on the findings would be beneficial. This could involve suggestions for long-term monitoring of heavy metal concentrations or studies on the effectiveness of remediation strategies

9. Ensure that the manuscript adheres to PLOS ONE's formatting guidelines, including reference style and figure/table formatting. Additionally, proofreading for grammatical errors and clarity will enhance the overall quality of the manuscript.

Reviewer #2: The abstract is longer than the introduction, and the introduction lacks a story to support the research study's clear objective.

The introduction should include the specific problem of soil facing the issue of heavy metal pollution.

A detailed overview of the problem, the research gap, the objectives, and the significance of your study should be discussed in the introduction.

Introduce Bahi district characteristics.

The spelling of anthropogenic is wrong in the introduction like anthroponenic*.

Lines 92-104 are copied and need to be rewritten

Lines 389-392 are copied and need to be re-written

Captions of the tables and figures need to be appropriate

In methodology, XRF used for metals composition report complete

Geospatial information of water samples is not well-defined

Table 1. Maximum standard permissible heavy metal concentrations in soil or water should be soil and water

Figure Captions are not well placed, and figure resolution is not up to the mark

Maps missing the proper labelling of sampling points

**Do you want your identity to be public for this peer review?** For information about this choice, including consent withdrawal, please see our Privacy Policy

Reviewer #1: No

Reviewer #2: No

---

## [Author Response · Author response to Decision Letter 1]

23 Dec 2024

PONE-D-24-37705 : Assessment of heavy metals in soil and water from Bahi district, Tanzania

RESPONSE TO JOURNAL REQUIREMENTS:

Item # Concern/response/action

Comment 1. Please ensure that your manuscript meets PLOS ONE's style requirements, including those for file naming. The PLOS ONE style templates can be found at

Action taken. PLOS ONE's style requirements observed

Comment 2: In your Methods section, please provide additional information regarding the permits you obtained for the work. Please ensure you have included the full name of the authority that approved the field site access and, if no permits were required, a brief statement explaining why.

Response: Included

Comment 3: We note that the grant information you provided in the ‘Funding Information’ and ‘Financial Disclosure’ sections do not match.

Response: Not clear, about the miss-match

Comment

Response: Not applicable

Comment 4. We note that your Data Availability Statement is currently as follows: All relevant data are within the manuscript and its Supporting Information files.

Response: All relevant data included in the manuscript or in form of supporting information

Comment

Action taken: Data available on manuscript or in form of supporting information

Comment

Response: Implemented

Comment

Response: Not applicable.

ADDITIONAL EDITOR COMMENTS:

Item # Concern/Response/Action taken

1. Methodology: Elaborate on sampling specifics, quality control, and statistical methods.

Response: Detailed included.

2.Data Visualization: Use graphical aids to make data interpretation more intuitive.

Response: Several figures were reproduced for clarity and by the use of SPACE

3. Discussion Expansion: Provide a thorough exploration of health and environmental implications, drawing connections with prior research.

Response: Detailed explanation of the results was considered. Results for each element were discussed

4. Language and Structure: Address grammatical and structural issues to meet journal standards, including reducing similarity and enhancing originality.

Response: To the level best of our comprehension we observed language issues.

RESPONSES TO REVIEWERS’ COMMENTS

Reviewer #1

The manuscript deals with "Assessment of heavy metals in soil and water from Bahi district, Tanzania" the following comments and suggestions are need to address before its consideration.

Comment 1: The methodology section should provide more detailed descriptions of the sampling process, including the specific locations and conditions under which samples were collected. This will help readers understand the context and reliability of the data

Response 1: Sampling Locations: We have specified the five villages (Bahi Makulu, Bahi Sokoni, Matajila, Chang’ombe, and Nagulo) where the soil and water samples were collected. These locations have been selected based on their proximity to Bahi Swamp and their significance in the district’s economic activities such as agriculture and fish-keeping.

Sampling Conditions: The manuscript now includes details about the sampling conditions, such as the time of collection, the weather conditions during sampling, and the tools and protocols used to prevent contamination during collection.

Coordinates: The exact GPS coordinates of each sampling site have been added to Supporting information to further improve the reproducibility of the study.

Comment 2: While the manuscript mentions the use of descriptive statistics, ANOVA, and PCA, it would be beneficial to elaborate on the rationale for choosing these specific statistical methods. Additionally, including a brief explanation of how these analyses contribute to understanding the relationships between heavy metals and metal oxides would strengthen the analysis section

Response 2: Comments implemented by revising the with inclusion of justification and explanation.

Comment 3: More emphasis on the quality control measures taken during the study, such as the use of blank solutions and recovery studies, would add credibility to the results. This could be integrated into the methodology section. https://link.springer.com/article/10.1007/s12011-024-04234-0,

https://www.sciencedirect.com/science/article/pii/S0278691524003727

Response 2: Familiarized with the content and context to improve the section

Comment 4 (1): What specific AAS techniques were utilized?

Response 4 (1): Specific AAS technique used was elaborated in -

Comment 4 (2): How do findings compare to previous studies?

Response 4 (2): Discussed table of comparison included

Comment 4 (3): What quality control measures were implemented? Referred to below papers and cite them

https://www.sciencedirect.com/science/article/pii/S0946672X24001019, https://www.sciencedirect.com/science/article/pii/S0946672X24000749,
https://www.sciencedirect.com/science/article/pii/S0889157523008177

Response 4 (3): Quality control measure explained.

Comment 5: The results section could be improved by including visual aids such as graphs or tables that summarize the heavy metal concentrations in both soil and water samples. This would make the data more accessible and easier to interpret for readers

Response 5: Visual aids improved by redrawing and few added in all case where applicable.

Comment 6: The discussion should delve deeper into the implications of the findings, particularly regarding the potential health risks associated with heavy metal concentrations. It would be useful to compare the results with similar studies in other regions to provide a broader context

Response 6: The discussion extended for every metal analyzed and compared with other from the literature.

Comment 7: The manuscript mentions the presence of radionuclides and heavy metals in the Bahi area. Expanding on how these factors interact with local agricultural practices and their potential impact on the community would provide valuable insights.

Response 7: To remain focused for the objective of this manuscript radionuclides were not discussed, however, overall consequence in highlighted.

Comment 8: Including a section that outlines potential future research directions based on the findings would be beneficial. This could involve suggestions for long-term monitoring of heavy metal concentrations or studies on the effectiveness of remediation strategies.

Response 8: Suggested as part of the recommendations.

Comment 9: Ensure that the manuscript adheres to PLOS ONE's formatting guidelines, including reference style and figure/table formatting. Additionally, proofreading for grammatical errors and clarity will enhance the overall quality of the manuscript.

Response 9: Comment addressed

Reviewer #2

The abstract is longer than the introduction, and the introduction lacks a story to support the research study's clear objective.

Comment 1: The abstract is longer than the introduction, and the introduction lacks a story to support the research study's clear objective.

Response 1: Harmonized to avoid losing the major focus of the manuscript.

Comment 2: The introduction should include the specific problem of soil facing the issue of heavy metal pollution.

Response 2: Elaborated to motive of undertaking the study foe levels of heavy metals was due to geological natural of Bahi district which characterized with the presence of uranium deposit and the co-existing of other toxic element was paramount.

Comment 3: A detailed overview of the problem, the research gap, the objectives, and the significance of your study should be discussed in the introduction.

Response 3: revised and exploited the context of the manuscript in the introduction section

Comment 4 : Introduce Bahi district characteristics.

Response 4 : Addressed in introduction and in study area section

Comment 5:The spelling of anthropogenic is wrong in the introduction like anthroponenic*.

Response 5: Corrected.

Comment 6: Lines 92-104 are copied and need to be rewritten

Response 6: Considered where necessary.

Comment 7: Lines 389-392 are copied and need to be re-written

Response 7: Considered.

Comment 8:Captions of the tables and figures need to be appropriate

Response 8: Revised as appropriate.

Comment 9: In methodology, XRF used for metals composition report complete

Response 9: Addressed with addition of technical details.

Comment 10: Geospatial information of water samples is not well-defined

Response 10: Geospatial information explained.

Comment 11: Table 1. Maximum standard permissible heavy metal concentrations in soil or water should be soil and water

Response 11:Corrected

Comment 12:Figure Captions are not well placed, and figure resolution is not up to the mark

Response 12:Addressed and some figures where reproduced.

Comment 13:Maps missing the proper labelling of sampling points

Response 13:Names or Wards where sampling was carried out are indicated.

---

## [Decision Letter · Decision Letter 1]

18 Mar 2025

Dear Dr. Sumary,

Thank you for submitting your manuscript to PLOS ONE. After careful consideration, we feel that it has merit but does not fully meet PLOS ONE’s publication criteria as it currently stands. Therefore, we invite you to submit a revised version of the manuscript that addresses the points raised during the review process.

We look forward to receiving your revised manuscript.

Kind regards,

Mohamed Y.M. Hanfi

Academic Editor

PLOS ONE

Reviewers' comments:

Reviewer's Responses to Questions

**Comments to the Author**

Reviewer #1: All comments have been addressed

Reviewer #2: All comments have been addressed

Reviewer #3: All comments have been addressed

2. Is the manuscript technically sound, and do the data support the conclusions?

Reviewer #1: Yes

Reviewer #2: No

Reviewer #3: Yes

3. Has the statistical analysis been performed appropriately and rigorously?

Reviewer #1: Yes

Reviewer #2: Yes

Reviewer #3: Yes

4. Have the authors made all data underlying the findings in their manuscript fully available?

Reviewer #1: Yes

Reviewer #2: Yes

Reviewer #3: Yes

5. Is the manuscript presented in an intelligible fashion and written in standard English?

Reviewer #1: Yes

Reviewer #2: Yes

Reviewer #3: Yes

**Reviewer #1:**  In my opinion, this new manuscript represents the authors' excellent work. The latest version of the manuscript has improved its quality.

**Reviewer #2:**  Manuscript lacks the strength in research hypothesis and objective. Research work is just based on very simple approach which do not reflect any potential outcome and reader preference as well as interest.

**Reviewer #3:**  Editor:

Please ensure the results are accurately reported, any overstated conclusions are rewritten and the limitations of the work fully explained.

Comments to the Authors

Dear Editor

PLOS ONE

Review report on manuscript #: PONE-D-24-37705R1

Title: “Assessment of heavy metals in soil and water from Bahi district, Tanzania”

The manuscript is significant, this study focuses on analyzing heavy metal concentrations in soil and water samples from Bahi district, Tanzania. The research utilized quantitative laboratory methods, specifically energy dispersive X-ray fluorescence (ED-XRF) spectrometry for soil samples and atomic absorption spectrometry (AAS) for water samples. Descriptive statistics, one-way ANOVA, Pearson Correlation Coefficient, and Principal Component Analysis (PCA) were employed to assess variations in heavy metal concentrations and their associations with metal oxides.

The introduction presents the specific work's aim in a well-organized manner. The experimental results were tabulated and illustrated in a comprehensive manner. Both abstracts and conclusions are informative and suitably formed.

After the corrections below, it can be suitable for the publication in the Journal.

Comment 1: The abstract is rather lengthy. A portion of the abstract ought to be moved into the Materials and Methods and Introduction sections.

Comment 2: The figures resolution of the article is very low especially fig. 3 to fig. 10. I couldn't see what is written on these figures.

Comment 3: Language and Style: The writing is mostly clear, but there are areas where grammatical improvements could enhance readability. A thorough proofreading is recommended.

Comment 4: The conclusion is poor. It should be rewritten according to the significance of the present outcomes in the context. The authors should mention how to achieve the research question mentioned at the end of the introduction. Conclusions must be deeper.

**Do you want your identity to be public for this peer review?** For information about this choice, including consent withdrawal, please see our Privacy Policy

Reviewer #1: No

Reviewer #2: No

Reviewer #3: **Yes: ** Sherif A. Taalab

---

## [Author Response · Author response to Decision Letter 2]

10 May 2025

RESPONSES TO REVIEWER’S COMMENTS.

Reviewer #1: In my opinion, this new manuscript represents the authors' excellent work. The latest version of the manuscript has improved its quality.

Response:

We appreciate for the reviewer’s time and comment for the work done.

Reviewer #2: Manuscript lacks the strength in research hypothesis and objective. Research work is just based on very simple approach which do not reflect any potential outcome and reader preference as well as interest.

Response

We appreciate the reviewer’s feedback on the research hypothesis and objectives. This study aims to address a gap in the literature on the geochemical distribution of heavy metals in Bahi District, a region with limited research, particularly concerning its uranium deposits. While the approach may seem straightforward, it is informed by the district's distinct geological and geochemical context.

The hypothesis posits that heavy metal concentrations in soils and water are influenced by both natural geological factors and anthropogenic activities, with potential health implications. The objectives were structured to evaluate concentration and distribution, providing baseline data essential for future studies and contributing to environmental chemistry and health research.

Regarding implementation scope, we acknowledge the study does not cover all variables but represents an essential step in understanding Bahi District’s environmental challenges. The findings not only address local concerns but are also relevant to similar semi-arid regions. This research establishes a foundation for future studies on bioaccessibility, source apportionment, and community engagement to mitigate long-term health risks.

We hope that this clarification addresses the reviewer’s concerns and enlightens the significance and potential impact of the study.

Reviewer #3: Editor:

Comment 1: The abstract is rather lengthy. A portion of the abstract ought to be moved into the Materials and Methods and Introduction sections.

Response:

o About 118 words were removed to reduce the length of the abstract while retaining the meaning and context.

The second and third sentence was removed since it also addressed in methodology (in study area section) and in the introduction section, respectively

Removed-About 96% of Bahi district is rural area, which the main economic activities are agriculture, fish-keeping and animal husbandry. The aim of this research was to assess the levels and spatial distribution of heavy metals in Bahi district. –

The sixth sentence was removed – it is well explained in the methodology part.

Removed- “Descriptive statistics; one-way analysis of variance (ANOVA), and Pearson Correlation Coefficient in conjunction with Principal Component Analysis (PCA) were used to determine variations in concentrations from different samples and the association between heavy metals or with metal oxides.”

The 10th sentence- shortened by deleting the ending phrase since it does not alter the meaning. Its relevance is captured in the conclusion section

Removed- “…showing non-uniform distribution and indicating various influencing factors.”

The third sentence (last but three) was removed. Since the re-phrasal of the last two serve the purpose as concluding phrases.

Removed- “These findings provide valuable insights into heavy metals distribution in the soils and waters sampled from Bahi localities, aiding in understanding environmental and health implications.”

Comment 2: The figures resolution of the article is very low especially fig. 3 to fig. 10. I couldn't see what is written on these figures.

Response:

o We decided to remove figs 4 to 10 and shift them to supporting documents as S6-S12 Figs. They can serve the purpose in case the reader would like to see the spatial distribution details. S1 Fig included in the list hence it is now S13 Fig in the list

o Hence the re-assignment of figure number was done Figs 11 – 16 are now identified as Figs 4 – 9, respectively.

Comment 3: Language and Style: The writing is mostly clear, but there are areas where grammatical improvements could enhance readability. A thorough proofreading is recommended.

Response:

o The proofreading has been undertaken to clear grammatical issues and typographic errors. This can be follow-up in the track-changes document.

Comment 4: The conclusion is poor. It should be rewritten according to the significance of the present outcomes in the context. The authors should mention how to achieve the research question mentioned at the end of the introduction. Conclusions must be deeper.

Response:

The conclusion has been re-written to reflect the following;

o Significance of outcomes: Highlighted how the findings address knowledge gaps on heavy metal distribution in Bahi District, emphasizing health risks linked to metal concentrations.

o Research question: Explicitly connected findings to the research question, focusing on metal concentrations and distribution

o Connection to health policies: Emphasized the integration of geochemical data with public health policy, offering actionable recommendations for mitigation and monitoring.

o How deeper is the conclusion? – For now, it highlighted the persistence of cadmium, toxicological concerns of tellurium, and manganese behavior, providing a deeper understanding of the findings.

---

## [Editor Report · Decision Letter 2]

15 May 2025

Assessment of heavy metals in soil and water from Bahi district, Tanzania

PONE-D-24-37705R2

Dear Dr. Dominic Parmena Sumary,

We’re pleased to inform you that your manuscript has been judged scientifically suitable for publication and will be formally accepted for publication once it meets all outstanding technical requirements.

Kind regards,

Mohamed Y.M. Hanfi

Academic Editor

PLOS ONE
---

## [Editor Report · Acceptance letter]

PONE-D-24-37705R2

PLOS ONE

Dear Dr. Sumary,

I'm pleased to inform you that your manuscript has been deemed suitable for publication in PLOS ONE. Congratulations! Your manuscript is now being handed over to our production team.

Kind regards,

on behalf of

Dr. Mohamed Y.M. Hanfi

Academic Editor

PLOS ONE